# Propofol induces mitochondrial-associated protein LRPPRC and protects mitochondria against hypoxia in cardiac cells

**Qianlu Zhang[1], Shiwei Cai[2], Liping Guo[3], Guojun Zhao[3]***

**1** Department of Anesthesiology, The First Affiliated Hospital of the University of South China, Hengyang, Hunan Province, P.R. China, **2** Cardiovascular Surgery, The First Affiliated Hospital of the University of South China, Hengyang, Hunan Province, P.R. China, **3** The Sixth Affiliated Hospital of Guangzhou Medical University, Qingyuan City People's Hospital, Qingyuan, Guangdong Province, P.R. China

* zzhcsu@163.com

## Abstract

**Data Availability Statement:** All relevant data are within the manuscript and its Supporting Information files.

### Background

Hypoxia-induced oxidative stress is one of the main mechanisms of myocardial injury, which frequently results in cardiomyocyte death and precipitates life-threatening heart failure. Propofol (2,6-diisopropylphenol), which is used to sedate patients during surgery, was shown to strongly affect the regulation of physiological processes, including hypoxia-induced oxidative stress. However, the exact mechanism is still unclear.

### Methods

Expression of LRPPRC, SLIRP, and Bcl-2 after propofol treatment was measured by RT-qPCR and western blot analyses. The effects of propofol under hypoxia were determine by assessing mitochondrial homeostasis and mitochondrial function, including the ATP level and mitochondrial mass. Autophagy/mitophagy was measured by detecting the presence of LC3B, and autophagosomes were observed by transmission microscopy

### Results

Propofol treatment inhibited cleaved caspase-9 and caspase-3, indicating its inhibitory roles in mitochondrial-related apoptosis. Propofol treatment also transcriptionally activated LRPPRC, a mitochondrial-associated protein that exerts multiple functions by maintaining mitochondrial homeostasis, in a manner dependent on the presence of hypoxia-induced factor (HIF)-1α transcriptional activity in H9C2 and primary rat cardiomyocytes. LRPPRC induced by propofol maintained the mitochondrial membrane potential (MMP) and promoted mitochondrial function, including ATP synthesis and transcriptional activity. Furthermore, LRPPRC induced by propofol contributes, at least partially, to the inhibition of apoptotic cell death induced by hypoxia.

**Funding:** This work was supported in part by research grants from the National Natural Science Foundation of China [81601103].

**Competing interests:** The authors have declared that no competing interests exist.

## Conclusion

Taken together, our results indicate that LRPPRC may have a protective antioxidant effect by maintaining mitochondrial homoeostasis induced by propofol and provide new insight into the protective mechanism of propofol against oxidative stress.

## Introduction

Under hypoxic conditions, cardiomyocyte hypertrophy is a process that meets the increased oxygen demand and maintains homeostasis [1]. Increased oxygen consumption induces oxidative stress and might induce physiological and even pathological events [2]. Thus, myocardial hypoxia is considered a major factor in cardiac ischemia and myocardial infarction and may cause microvascular disease [3] and heart failure [4]. Since cardiomyocytes are terminally differentiated cells that are not believed to regenerate, preventing the cardiomyocyte injury and loss induced by hypoxia is a vital therapeutic strategy. Thus, targeting hypoxic signaling is considered a promising strategy to protect against heart injury [5].

Propofol is a widely employed intravenous anesthetic that has also been found to display protective effects on oxidative stress in different types of cells, including cardiomyocytes [6], hippocampal neurons [7], and kidney cells [8]. It has been reported that in rat cardiac H9c2 cells, propofol transcriptionally activates antioxidant enzymes to abolish oxidative stress induced by $H_2O_2$ [6]. Li and colleagues found that hypoxia-induced oxidative stress was decreased after inhibition of calcineurin-induced calcium overload and activation of the subsequent YAP signaling pathway in hippocampal neurons [6]. Propofol treatment was also proven to exert a protective effect on kidney damage after sepsis-induced AKI by inhibiting oxidative stress [9]. However, given that oxidative stress is one of the most critical factors of mitochondrial damage [10], the exact effect of propofol on oxidative stress-induced mitochondrial damage is still largely unknown.

Leucine-rich pentatricopeptide repeat-containing (LRPPRC), also known as LRP130, is a member of the pentatricopeptide repeat protein family and has multiple functions in various processes, including homeostasis, microtubule alterations, RNA stability, DNA/RNA binding, transcriptional activity in the mitochondria, metabolic processes, RNA nuclear export, tumorigenesis and progression [11–15]. LRPPRC is essential for maintaining mitochondrial homeostasis by protecting the mitochondrial membrane through interactions with Beclin 1 and Bcl-2 and formation of a ternary complex to maintain Bcl-2 stability [16,17,18]. This protein was also found in the mitochondrial matrix and binds to single-stranded RNA to post-transcriptionally regulate mitochondrial genes [19]. LRPPRC acts as an inhibitor of autophagy/mitophagy by maintaining MMP and thus promoting mitochondrial function [17].

SRA stem-loop interacting RNA binding protein (SLIRP) was initially isolated as a protein that binds to the stem-loop structure of the RNA molecule SRA [20]. An RNA recognition motif (RRM) domain of SLIRP was shown to physically bind to pentatricopeptide repeat (PPR) domains on LRPPRC and thus regulates mitochondrial protein synthesis by protecting LRPPRC from degradation [21–23]. Interestingly, SLIRP was also found to be a novel protein that interacted with Bcl-2 [23]. Although SLIRP was not found to mediate the ability of Bcl-2 to protect against apoptosis and oxidative damage, the binding of Bcl-2 to SLIRP stabilizes the SLIRP protein and regulates mitochondrial mRNA levels. This finding raised the question of whether LRPPRC, SLIRP and Bcl-2 form heterotrimers and function synergistically. These results could provide new insight into the regulation of these proteins on cell physiological processes and mitochondrial function.

In this study, we established a model of hypoxia in H9c2 cells and primary rat cardiomyocytes to investigate the effects of propofol on hypoxia-induced myocardial damage. We found that propofol exerts protective effects against hypoxia-induced apoptosis potentially by upregulating LRPPRC. LRPPRC upregulation attenuated the hypoxia-induced cardiomyocyte injury by maintaining mitochondrial homeostasis and function, suggesting a novel protective mechanism of propofol against hypoxia-related heart diseases.

## Material and methods

### Cell culture and treatment

H9C2 cell line was originally derived from the embryonic rat ventricle and was bought from American Type Culture Collection (ATCC Manassas, VA, USA). Cells were stored in liquid nitrogen and cultured in Dulbecco's modified Eagle's medium (DMEM) supplemented with 10% Fetal bovine serum (FBS, Life Technologies, Grand Island, NY, USA), 100 U/mL penicillin and 100 g/mL streptomycin in a humidified atmosphere of 5% $CO_2$ and 95% air at 37˚C.

Hypoxic cultures were carried out in a humidified hypoxia workstation invivo 400 model (Ruskin Technology Ltd, Bridgend, UK). Briefly, medium was pre-exposed to 5% $CO_2$, 20% (Normoxia) or 1%$O_2$ (Hypoxia) balanced with nitrogen. Then, cells were cultured in pretreated medium and kept in corresponding conditions for 24 or 48 h.

### Animals and isolation of cardiac cells

2-day-old Wistar rats were used for isolating primary rat cardiac myocytes (RCM). All of the experiments were performed in accordance with the Guide for the Care and Use of Laboratory Animals and approved by the First Affiliated Hospital of the University of South China ethics committee.

The hearts were dissociated into single suspended cells by using SoniConvert® Sonicator (UTL: http://www.doc-sense.com/index.html, DocSense, Chengdu, China) following the manufacturer's instruction. Briefly, the hearts were rapidly excised and rinsed in ice-cold PBS. The hearts were minced and incubated in DS buffer supplied by DocSense (Chengdu, China), and incubated at 37˚C for 5 min with several times of rotation. Then Sonicator was employed to dissociate tissue into single cells with the procedure: 3 s under 10% power. After a brief centrifugation at 400 g, 4˚C for 5 min, cell pellet was collected for further culturing and analysis. For culturing, cells were transferred to serum-free Cardiac Myocyte Medium (ScienCell Research Laboratories, cat. No.: 6101, Carlsbad, CA). Cells were maintained in a $CO_2$ incubator at 37˚C in humid air containing 5% $CO_2$. The medium was refreshed every two days.

### Annexin V/PI double staining

Cultured cells were suspended using 0.25% Trypsin and washed with ice-cold PBS twice to remove trypsin and EDTA. Then cells were suspended in ice-cold PBS and adjust the cell concentration to $1\times10^6$ cells per 1mL. The cells were then stained with Annexin V-FITC (green fluorescence) and PI (red fluorescence), which allowed the identification of intact cells (FITC-/PI-), early stage of apoptotic cells (FITC+/PI), late stage of apoptotic cells (FITC+/PI+) and unapoptotic cells (FITC-/PI+). Then stained cells were analyzed by 3 laser Navios flow cytometers (Beckman Coulter, Brea, CA, USA). The data files was analyzed using FlowJo 9.7.6 (FlowJo LLC, Ashland, OR USA).

### JC-1 staining

JC-1 probe was employed to investigate the mitochondrial membrane potential to determine the mitochondrial homeostasis. JC-1 (20 μg/mL, Life Technologies, Grand Island, NY, USA)

was added and incubated at 37˚C for 20 min and then washed twice with ice-cold PBS to remove remained dye. Images were taken under a X71 (U-RFL-T) fluorescence microscope (Olympus, Melville, NY). The red fluorescence emission maximum at 590 nm represents JC-1 aggregates, and green fluorescence emission maximum at 529 nm represents JC-1 monomers.

## siRNA transfection

Knockdown of mRNA level of *LRPPRC* was achieved by transient transfection of cells with siRNA duplexes (Thermo Scientific, Waltham, MA, USA), specific to the mRNAs of *LRPPRC*. The relevant siRNA sequences were: 5′-GCCUGCCGAUUGAACCAAATT-3`and 5′-UUUGG UU CAAUCGGCAGGCAA-3′; negative control (NC) sense 5′-GUUCAAUAUUAUCAAGCGG UU-3′ and antisense 5′-CCGCUUGAUAAUAUUGAACUU-3′. According to author's instruction, $2\times10^5$ cells were grown in 2 ml of serum-free medium. A siRNA/transfection reagent complex was formed at room temperature by combining siRNA oligomer (50 nM) with 5 μl (2 μg/ml) Lipofectamine™ 2000 transfection reagent (Thermo Scientific, Waltham, MA, USA) in 0.5 ml OptiMEM medium (Thermo Scientific, Waltham, MA, USA), and this was applied to cells for 48 h until they were harvested. Cells transfected with NC siRNA was considered as Control cells.

## Real-time quantitative PCR (RT-qPCR)

To detect mitochondrial transcriptional activity, the transcriptional levels of mitochondrial-coded genes, including COX I and ND1, were analyzed by RT-qPCR. $1\times10^6$ cells were collected by trypsin and pelleted by centrifugation at 1000g for 5 min. 1 ml of TRIzol reagent (Life Technologies, Grand Island, NY, US) was added and pellet was sonicated by SoniConvert™ system (DocSense, Chengdu, China) and total RNA was extracted according to the manual. Then isolated RNA was reverse transcribed using a Reverse Transcriptase Kit (RIBO-BIO, Guangzhou, China). The PCR was performed using Power SYBR™ Green PCR Master Mix Kit (Applied Biosystems, Foster City, CA, USA). The qPCR was performed in a ABI7500 system (Applied Biosystems, Foster City, CA, USA) under the following conditions: 95˚C 5 min, 35 cycles of 95˚C 10s, and 60˚C 1min. The specific primers used were followed: LRPPRC, 5′-CTGCACTGTGCTCTTCAAGC-3′ and 5′-GACTGCACACTACCGAAGCA-3′; COX I forward: 5′- GGAGCAGTATTCGCCATCAT-3′; reverse: 5′-GAGCACTTCTCGTTTTGAT GC-3′; ND1 forward: 5′- CACCCCCTTATCAACCTCAA-3′; reverse: 5′-ATTTGTTTC TGCGAGGGTTG-3′; GAPDH forward: 5′-CCTTCATTGACCTCAACTACAT-3′; reverse: 5′-CCAAAGTTGTCATGGATGACC-3′.

To detect mitochondrial DNA contents, total DNA was isolated using Genome DNA isolation kit (DocSense, Chengdu, China) and 30 ng of DNA was used as template for each reaction. The same reacting system and running condition was the same with RT-qPCR. The specific primers used were followed: 12S rDNA forward: 5′- ACCGCGGTCATACGATTAAC-3′; reverse: 5′- AGTACCGCCAAGTCCTTTGA-3′; 18S rDNA forward: 5′-TCAATCTC GGGTGGCTGAACG-3′; reverse: 5′-GGACCAGAGCGAAAGCATTTG-3′.

## Western blot

The primary antibodies used was listed as followed: Rabbit monoclonal anti-**LRPPRC** antibody (1: 2000, #ab97505); Rabbit monoclonal anti- **Bcl-2** antibody (1:1000, #abab32124); Rabbit monoclonal anti- **Beclin 1** antibody (1:1000, #ab208612); rabbit monoclonal anti-cleaved PARP1 antibody (1:2000, #ab32561); rabbit monoclonal anti-pro-caspase-3 antibody (1:2000, #ab32150); rabbit monoclonal anti-cleaved-caspase-3 antibody (1:1000, #ab49822); rabbit monoclonal anti-pro caspase-8 antibody (1:1000, #ab108333); mouse monoclonal anti-

cleaved-caspase-8 antibody (1:1000, #9496S, CST); mouse monoclonal anti-caspase-9 antibody (1:1000, #9508T, CST). Rabbit monoclonal anti-β-actin antibody (1:5000, #ab8227). Goat anti-rabbit IgG H&L antibody (HRP ladled, 1:10000, #ab7090) was used as secondary antibody. Blot bands were quantified via densitometry with Image J software (National Institutes of Health Baltimore, MD, USA). β-actin was used as an internal reference.

## Immunoprecipitation

For immunoprecipitation assay, $5×10^6$ cells were lyzed using SoniConvert® sonicator (Doc-Sense, Chengdu, Sichuan, China. URL: http://www.doc-sense.com/index.html) according to the manufacturer's instruction. 200 μl of supernatant was incubated with 10 μg of Anti-Flag antibody, Anti-HA antibody or rabbit IgG (negative control) followed by immunoprecipita-tion with 50 ul of protein A agarose beads during and overnight incubation at 4˚C with rota-tion. Product was denatured in 200 μl of 1×SDS loading buffer and immunoblotting was performed as described before.

## Mitochondrial ATP synthesis

To measure ATP synthesis, cells were suspended in solution containing 0.22M sucrose, 0.12 M mannitol, 40 mM Tricine, pH7.5, and 1mM EDTA and lysed using SoniConvert$^{TM}$ system and analyzed using Optocomp I BG-1 luminometer (GEM Biomedical Inc.) using the ATP Bioluminescent Assay kit (Sigma).

## Statistical analysis

All data were analyzed for statistical significance using SPSS 13.0 software (SPSS, Chicago, IL, USA) and presented as means±SD from at least 3 independent experiments performed in duplicate. Statistical comparisons of the results were made using analysis of One-way ANOVA. $P < 0.05$ was considered statistically significant.

# Results

## Propofol exerts a protective effect against hypoxia-induced cell injury

To determine the effect of propofol on hypoxia-induced cell injury, we subjected the cells to hypoxic conditions for 0, 24 and 48 h with different concentrations of propofol ranging from 10 to 100 mM, and cell viability was assayed using the CCK-8 assay. As shown in Fig 1A, 25, 50 and 100 mM propofol inhibited the decrease in viability after 48 h of treatment normalized to normoxia group ($P < 0.05$). Treatment with 50 mM propofol was further used to detect hyp-oxia-induced apoptotic cell death after 48 h. As expected, both early- and late-stage apoptosis induced by hypoxia decreased significantly after propofol treatment ($P < 0.05$, Fig 1B). We then investigated PARP, pro-caspase 3, cleaved-caspase 3, pro-caspase-9, cleaved-caspase 9 and pro-caspase 8 and observed increased protein levels of cleaved-caspase-3, cleaved-caspase 9, and cleaved-PARP after hypoxia exposure; these changes were significantly reversed by the addition of propofol ($P < 0.05$, Fig 1C). Consistently, propofol also decreased the caspase 3 and caspase 9 activities, which were stimulated by hypoxia exposure ($P < 0.05$, Fig 1D).

## Propofol transcriptionally activates LRPPRC under hypoxic but not normoxic conditions

To determine whether LRPPRC and its related proteins are involved in the effects of propofol, we detected LRPPRC, Bcl-2 and Beclin 1. As expected, LRPPRC increased after 24 h and 48 h of incubation with propofol under hypoxia ($P < 0.05$, Fig 2A). Furthermore, Bcl-2 showed a

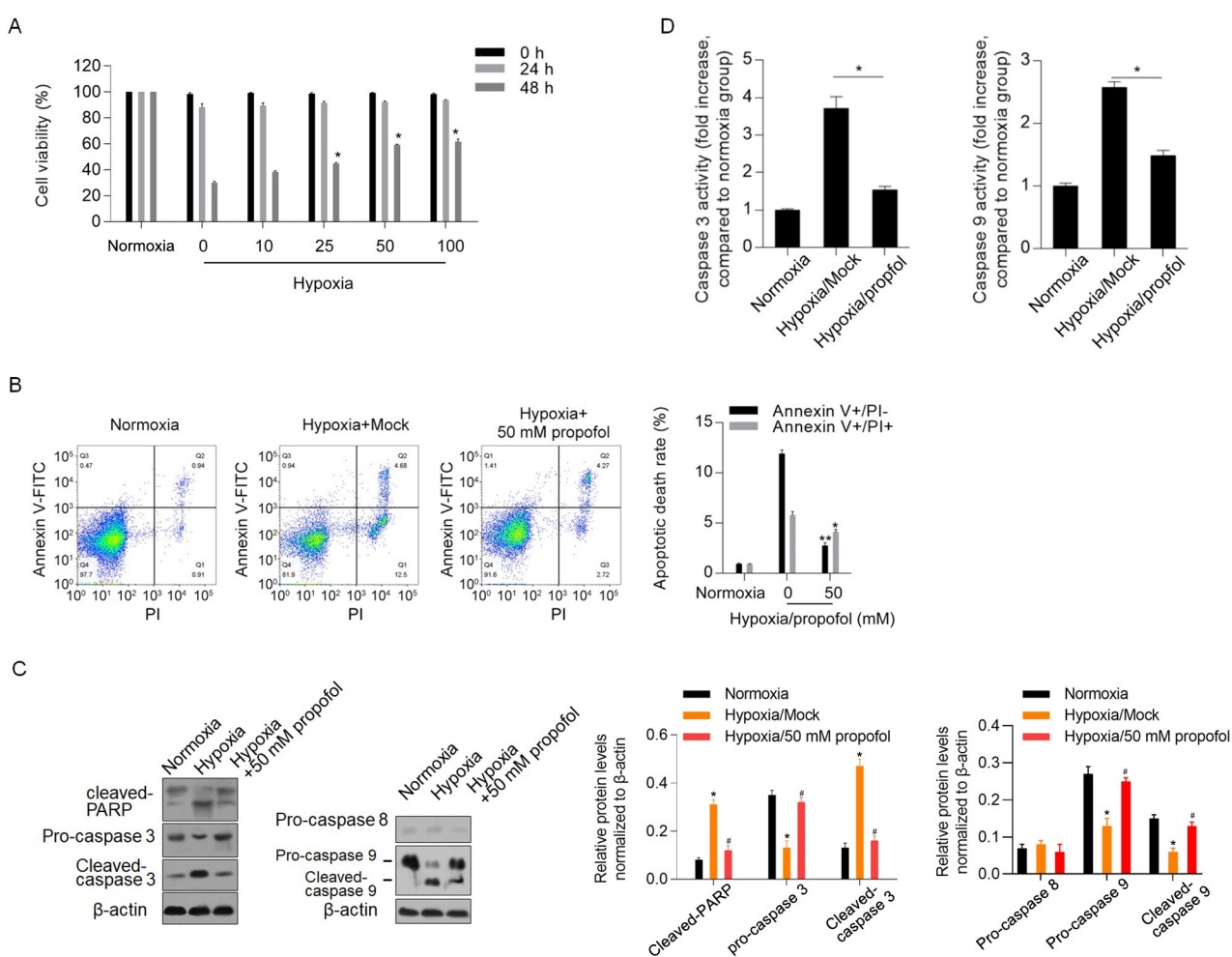

**Fig 1. Propofol exerts protective effect against hypoxia-induced cell injury via inhibiting mitochondria-related apoptosis.** (A) The cell viability was assayed using CCK-8 when H9c2 cardiomyocytes were incubated with 1% O2 after 0, 24 and 48 h. (B) Cell apoptosis was detected using Annexin V-FITC/PI double staining after being incubated with 1% O2 with or without 50 mM of propofol for 48 h. (C) the protein level of PARP, pro-caspase 3, pro-caspase 8, pro-caspase 9, cleaved-caspase 3 and cleaved-caspase 9 were detected using western blot. $^*P<0.05$, vs. Normoxia group; $^\#P<0.05$, Hypoxia/Mock group. (D) caspase 3 and caspase 9 activity was measured. $^*P<0.05$, vs. Normoxia group; $^\#P<0.05$, Hypoxia/Mock group.

similar trend as LRPPRC. RT-qPCR analysis further confirmed the transcriptional activation of LRPPRC (P<0.05, Fig 2B). We then performed immunofluorescence staining to localize LRPPRC and found that propofol-induced LRPPRC localized around the nucleus (Fig 2C). Notably, propofol treatment under normoxic conditions failed to affect both the mRNA and protein levels of LRPPRC (Fig 2D), indicating that propofol regulates LRPPRC in the context of hypoxia biology.

## Propofol transcriptionally activates LRPPRC depending on the transcriptional activity of HIF-1α

To detect the possible mechanism of upregulation of LRPPRC by propofol, we used hypoxia, the hypoxia mimic DFO and CoCl₂ for treatment of H9c2 cells. As shown in Fig 3A, hypoxia, DFO and CoCl₂ cotreatment with propofol all significantly increased LRPPRC, and this change was reversed by the addition of a transcriptional inhibitor of HIF-1α, 1 ng/ml echino-mycin. RT-qPCR showed that propofol transcriptionally activated LRPPRC under oxidative

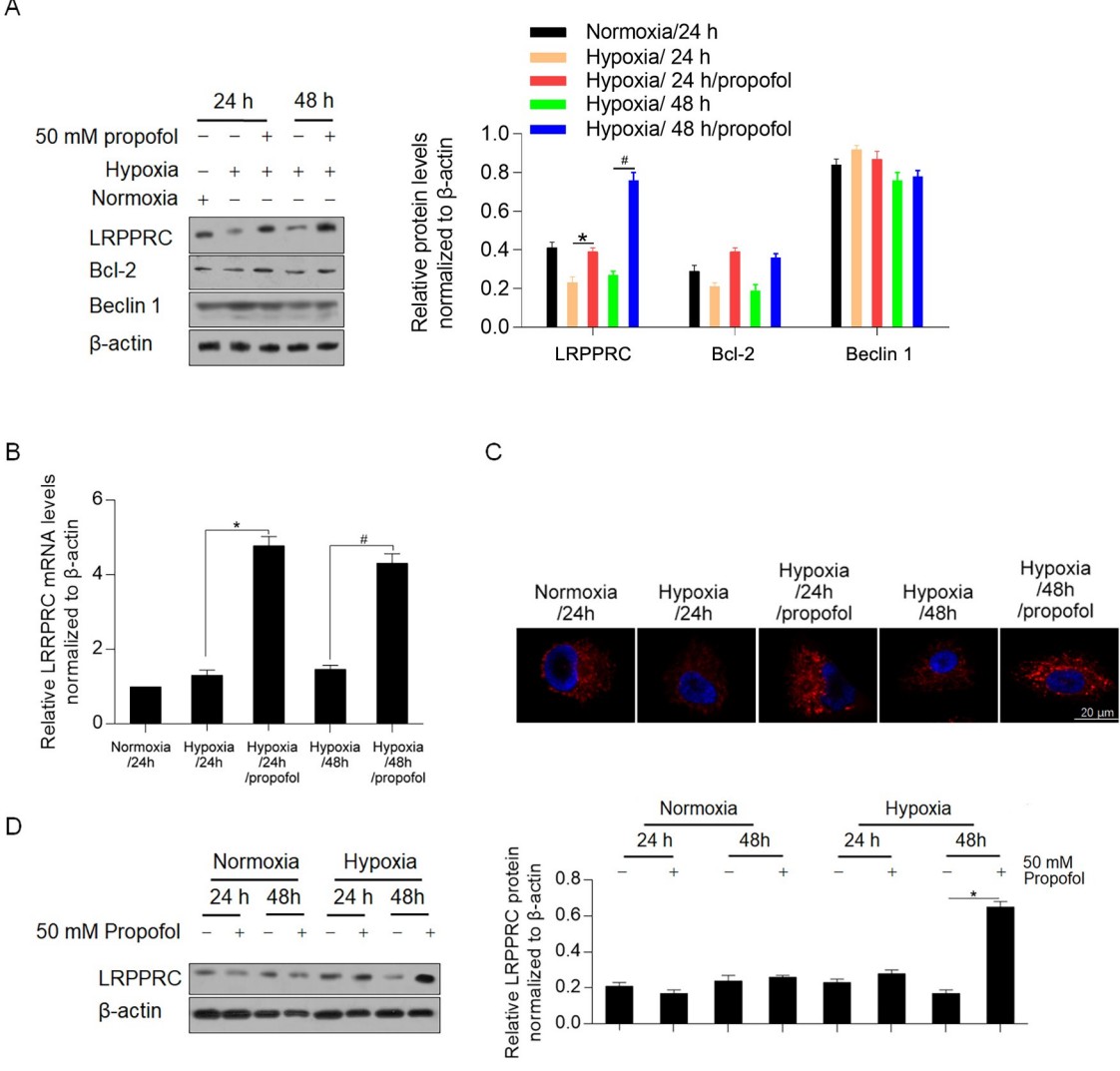

**Fig 2. Propofol transcriptionally induced LRPPRC under hypoxic condition.** (A) western blot was performed to detect the effects of propofol on LRPPRC, Bcl-2 and Beclin 1 protein levels under hypoxic condition. $^*$P<0.05, vs. Hypoxia/24 h group. $^\#$P<0.05, Hypoxia/48 h group. (B) RT-qPCR assay was performed to detect the mRNA level of LRPPRC after propofol exposure. (C) immunofluorescent staining was performed to localize LRPPRC in H9c2 cells. (D) RT-qPCR assay was performed to detect the effect of propofol under normoxic condition.

stress (Fig 3B). All these results demonstrated that propofol transcriptionally activated LRPPRC in the presence of HIF-1α transcriptional activity. Analysis of apoptotic death demonstrated consistent results, showing that the antiapoptotic effect of propofol was reversed by the addition of echinomycin (Fig 3C).

To further confirm whether propofol regulates LRPPRC in primary cardiomyocytes, neonate cardiomyocytes were chosen for further confirmation, instead of adult cardiomyocytes for the poor survival period and low transfection efficacy. Firstly, we isolated RCM from 2-day-old rat and cultured in 6-well plate (Fig 4A). After normoxia, hypoxia and hypoxa/propofol treatment for 24 h, consistently, LRPPRC was immune-stained and presented to be uprelated in hypxia/propofol co-treated group (Fig 4B). Hypoxia exposure, DFO and CoCl$_2$ treatment obviously upregulated HIF-1α protein (Fig 4C), and upregulated LRPPRC protein with the presence of

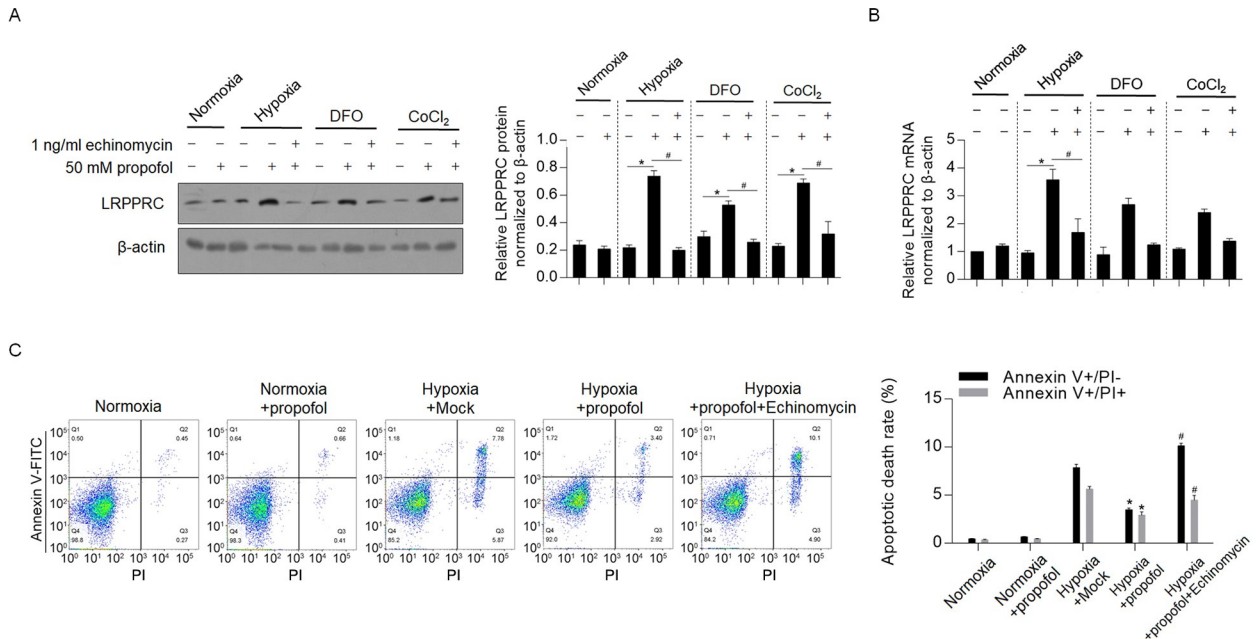

**Fig 3. Propofol transcriptionally activated LRPPRC.** (A) Western blot was employed to detect LRPPRC protein after hypoxia, DFO and CoCl2 treatment. (B) RT-qPCR assay was employed to detect mRNA level of LRPPRC. (C) Cell treated with propofol under hypoxia was analyzed for apoptotic death rate.

propofol, which were reversed by inhibiting HIF-1α transcriptional activity, indicated the consistent finding which was observed in H9C2 cells. We then also detect the effect of propofol under hypoxia on apoptosis in primary cardiomyocytes. As it is shown in Fig 4D, addition of propofol inhibited hypoxia-induced apoptosis, which was reversed by inhibiting HIF-1α transcriptional activity. All these results suggested that, propofol exerts protective effect against hypoxia-induced apoptosis and upregulated LRPPRC via HIF-1α transcriptional activity.

## Propofol maintains the mitochondrial membrane potential and mitochondrial functions under hypoxic conditions depending on the activation of LRPPRC

After exposure to hypoxia for 24 h with or without propofol, JC-1 staining was performed, and we observed that the decrease in aggregates induced by hypoxia exposure was reversed by the addition of propofol, and this change was abolished by inhibition of HIF-1α transcriptional activity (Fig 5A). As expected, propofol also promoted ATP synthesis (Fig 5B), mitochondrial DNA content (Fig 5C) and transcriptional activity (Fig 5D), demonstrating that propofol protects mitochondria from hypoxia-induced mitochondrial dysfunction. Furthermore, we detected the status of autophagy after hypoxic exposure. As shown in Fig 5E, the LC3-II level was decreased at both 12 and 24 h after the addition of propofol, indicating that propofol also potentially inhibited hypoxia-induced autophagy and thus inhibited mitochondria degradation induced by membrane damage. We then = observed autophagic vacuoles affected by LRRPRC. As shown in Fig 4F, in the presence of the lysosomal inhibitor bafilomycin A1, propofol treatment decreased the number of autophagic vacuoles, which was reversed by the addition of echinomycin (Fig 5F).

We further determined whether propofol-induced LRPPRC is critical for propofol's protective effect against hypoxia. siRNA targeting LRPPRC mRNA (siLRPPRC) was introduced

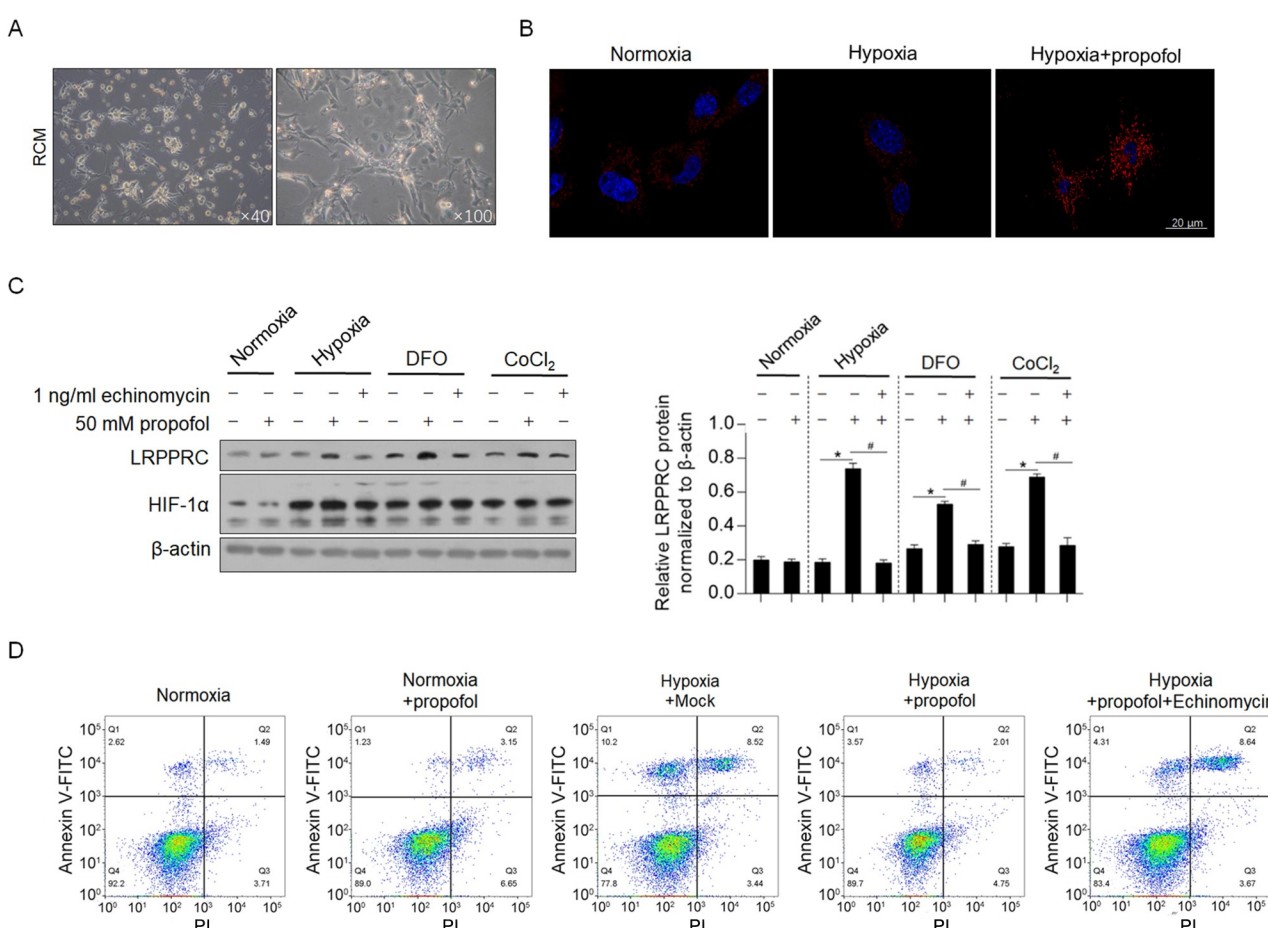

**Fig 4. Propofol exerts protective effect against hypoxia-induced cell apoptosis in RCM.** (A) Isolated RCM was cultured. (B) immunofluorescent staining was performed to localize LRPPRC in RCM. (C) The LRPPRC and HIF-1α protein levels were detected by western blot. (D) Cell treated with propofol under hypoxia was analyzed for apoptotic death rate.

before hypoxia exposure, and western blot analysis showed that siLRPPRC efficiently knocked down endogenous LRPPRC under normoxia and upregulated LRPPRC under hypoxia (Fig 6A). After LRPPRC knockdown, the ratio of LC3B-II/LC3B-I increased compared to that of the control group (siNC transfected), indicating that endogenous LRPPRC inhibited the basal level of mitophagy/autophagy (Fig 6A). As expected, under hypoxic conditions, propofol reversed the increased ratio of LC3B-II/LC3B-I, which was abolished by LRPPRC knockdown (Fig 6A). We detected ATP synthesis and mitochondrial DNA mass and consistently observed that knockdown of LRPPRC under both normoxia and hypoxia decreased ATP synthesis and mitochondrial DNA mass, demonstrating that LRPPRC may be critical for maintaining mitochondrial function (Fig 6B&6C).

## Hypoxia-induced LRPPRC stabilizes Bcl-2 potentially by binding to it

To confirm whether hypoxia regulates Bcl-2 via LRPPRC induction, we treated the hypoxia-exposed cells with propofol after LRPPRC knockdown. As expected, propofol treatment obviously increased LRPPRC under hypoxia exposure, and LRPPRC knockdown obviously decreased the Bcl-2 protein level compared to that in the hypoxia/mock group (Fig 7A). Moreover, Beclin 1 was not obviously affected by upregulated LRPPRC. As LRPPRC tightly

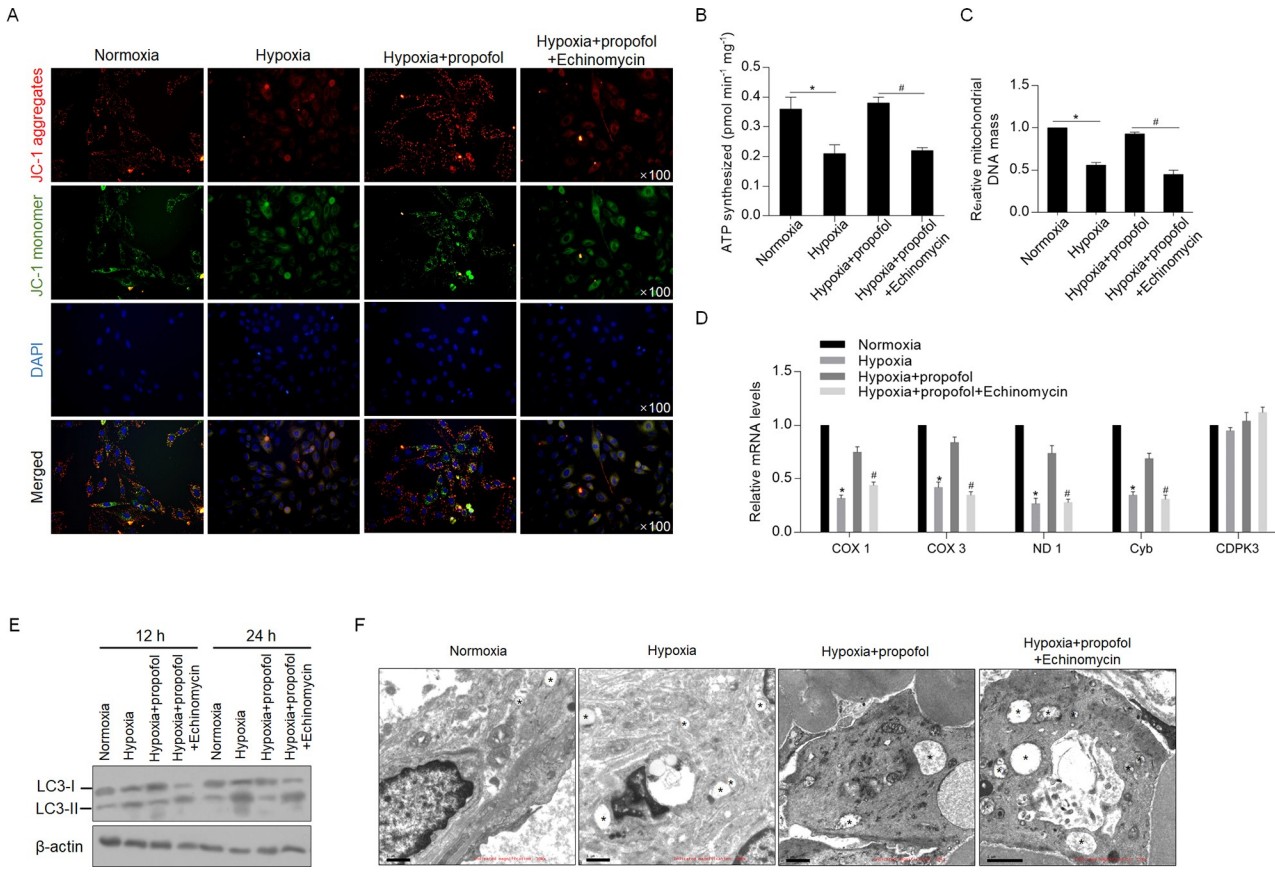

**Fig 5. Propofol contributes to maintenance of mitochondrial homeostasis and mitochondrial functions.** (A) H9c2 cells exposed to hypoxia with or without propofol were stained with JC-1 to identify mitochondrial membrane potential. (Red, JC-1 aggregates; green, JC-1 monomer). The effects of propofol on regulating ATP synthesis (B), mitochondrial DNA mass (C), and transcriptional activity (D) were measured. (E) western blot was performed to detect LC-3 level. (F) Transmission electron microscopy imaging of cells exposed hypoxia with or without propofol. *, autophagic vacuoles.

regulates the Bcl-2 protein stability by binding directly to it, we wanted to determine whether propofol-induced LRPPRC protein enhances binding to Bcl-2 [16–18]. After immunoprecipitation, the obtained products were detected for the existence of LRPPRC, Bcl-2, Beclin 1 and β-actin. The intensive signal of LRPPRC and undetectable β-actin demonstrated the specific and efficient precipitation of LRPPRC by an anti-LRPPRC antibody (Fig 7B). Notably, the most intensive signals of LRPPRC, Bcl-2 and Beclin 1 were observed after propofol treatment, and LRPPRC knockdown decreased Bcl-2 and Beclin 1 in the IP product. These results indicated that propofol-upregulated LRPPRC binds more strongly to Bcl-2 and Beclin 1 and may affect the stability of Bcl-2 but not Beclin 1.

## Hypoxia-induced LRPPRC binds the SLIRP protein

Given that SLIRP was reported to bind to and prevent the degradation of LRPPRC and also interact with Bcl-2 [21–23], we assessed whether SLIRP is affected under hypoxic conditions. Western blotting showed that SLIRP had a similar trend to LRPPRC at the protein level (Fig 8A), without being affected transcriptionally. The direct binding of SLIRP to LRPPRC or Bcl-2 is necessary for its stabilizing effects on LRPPRC or Bcl-2; thus, we tried to determine whether SLIRP, LRPPRC and Bcl-2 form heterotrimers. We tried to precipitate endogenous SLIRP or

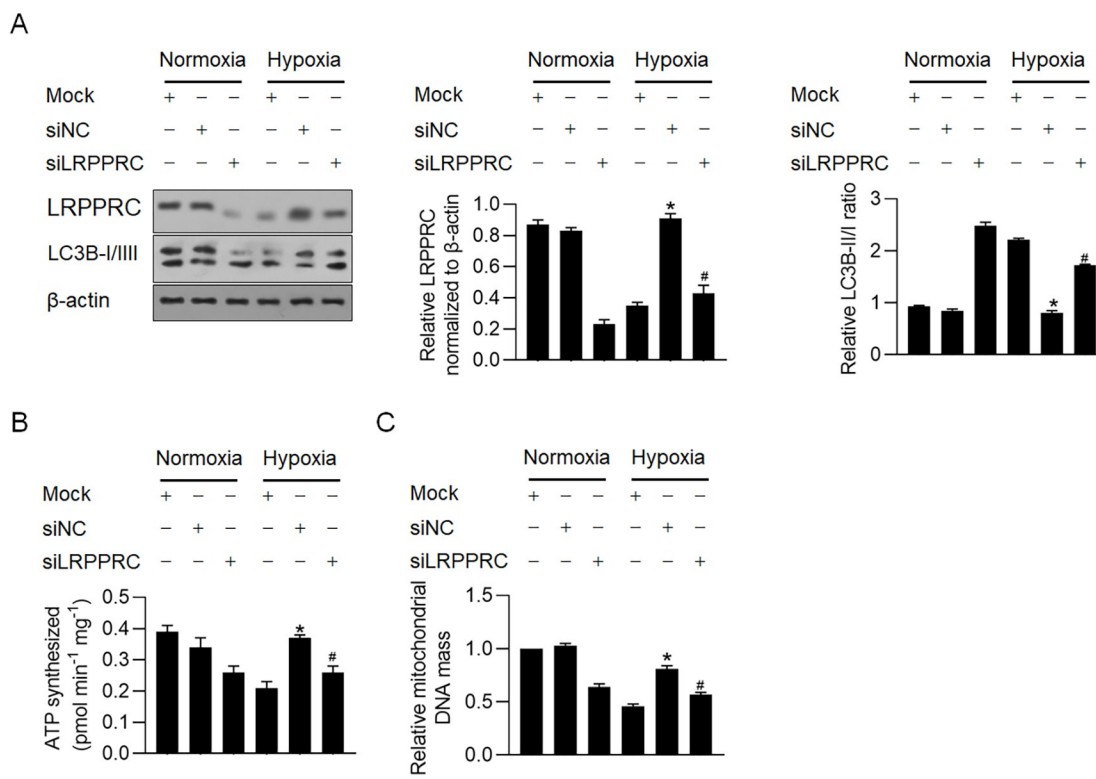

**Fig 6. Propofol potentially inhibited autophagy/mitophagy and maintained mitochondrial function via upregulated LRPPRC.** (A) After efficiently knockdown of LRPPRC, LC3B-I/II level was detected by western blot. *P<0.05, vs. hypoxia/mock group. #P<0.05, vs. hypoxia/siNC group. After hypoxia exposure with or without LRPPRC knockdown, ATP synthesis (B) and mitochondrial DNA mass (C) were measured. *P<0.05, vs. hypoxia/mock group. #P<0.05, vs. hypoxia/siNC group.

LRPPRC. However, SLIRP and LRPPRC interactions were not observed, which may be due to low endogenous protein levels. Thus, we overexpressed Flag-tagged LRPPRC and HA-tagged SLIRP in H9C2 cells and reperformed the immunoprecipitation analyses. As presented in Fig 8C (right panel), Flag-LRPPRC and HA-SLIRP were intensively expressed after enrichment. Flag-LRPPRC and HA-SLIRP were successfully precipitated, and in separate IP products, the LRPPRC, SLIRP and Bcl-2 proteins were all detected (Fig 8C), indicating that LRPPRC, SLIRP and Bcl-2 potentially form heterotrimers and thus function synergistically.

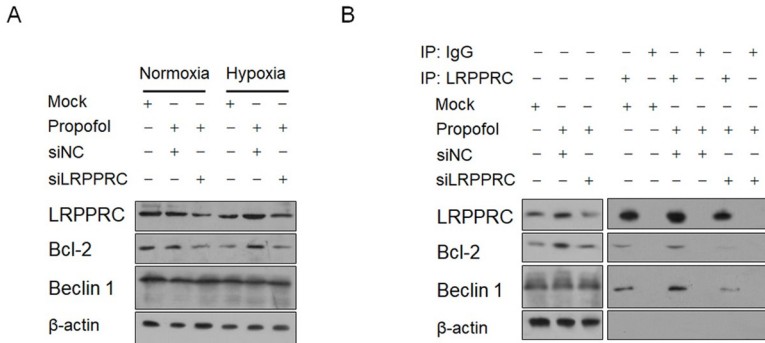

**Fig 7. Propofol-induced LRPPRC physically interacted with Bcl-2 and Beclin 1.** (A) LRPPRC knockdown using siRNA transfection obviously decreased Bcl-2 protein. (B) Immunoprecipitation was carried out to detect the physical interaction between LRPPRC, Bcl-2 and Beclin 1.

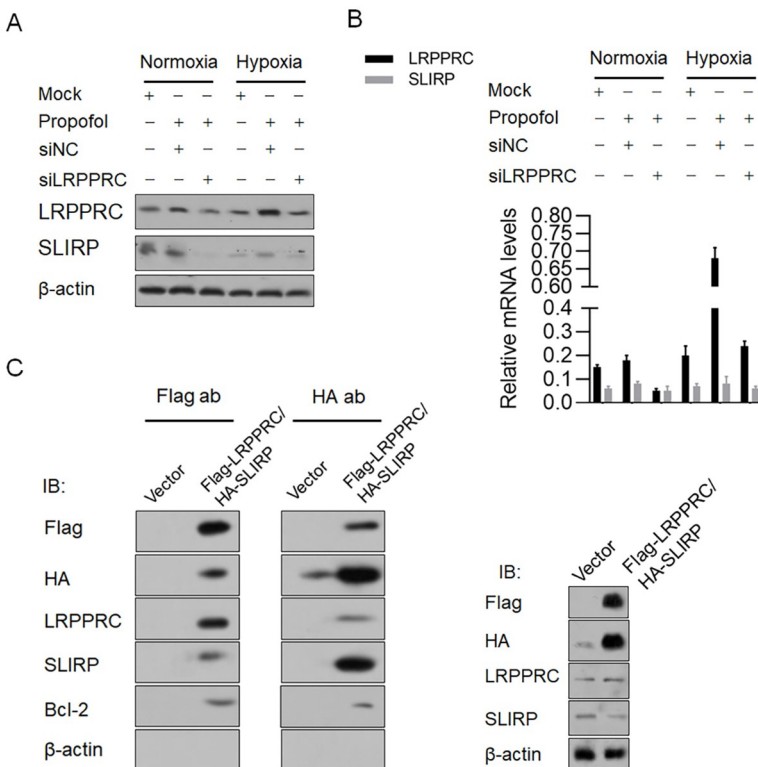

**Fig 8. Propofol-induced LRPPRC interacted with SLIRP.** (A) Propofol treatment affected SLIRP protein level under hypoxia. (B) SLIRP mRNA level was measured by RT-qPCR after propofol treatment under hypoxia. (C) immunoprecipitation was carried out to detect the potential formation of heterotrimer of LRPPRC, SLIRP and Bcl-2.

## Discussion

In this study, we demonstrated the protective effects of propofol on cardiomyocyte injury induced by hypoxia. Propofol maintained mitochondrial homeostasis and promoted mitochondrial function. In rat H9C2 cells, propofol reversed apoptotic cell death induced by oxidative stress, indicating its potential protective mechanism by protecting mitochondria.

Propofol has critical antioxidant and antiapoptotic effects [24]. In human umbilical vein endothelial cells (HUVECs) exposed to $H_2O_2$-induced oxidative stress, propofol abolished reactive oxygen species and thus inhibited apoptotic cell death. Propofol attenuated the cleavage of caspase-3 and the expression of Bcl-2 and Bax in a dose-dependent manner. This drug also ameliorated $H_2O_2$-induced phosphorylation of the p38 MAPK, JNK, and Akt signaling pathways, which are critical for initiating or promoting apoptosis [25]. Here, we revealed that propofol treatment protected H9c2 cells from hypoxia-induced cell apoptosis. We further found that propofol treatment substantially increased mitochondrial function, including increasing ATP synthesis, mitochondrial DNA content and transcriptional activity. Mitochondria play key roles in inducing apoptotic cell death via ROS-inducing pathways [26], including regulating calcium levels and opening of the mitochondrial permeability transition pore followed by release of cytochrome c. Sumi and colleagues reported that propofol treatment targets mitochondrial complexes I, II, and III and induces a cellular metabolic switch from oxidative phosphorylation to glycolysis [26]. Subsequently, accumulated ROS activated caspase 9 and 3/7 and induced mitochondrial-dependent apoptosis. According to our data, propofol treatment decreased mitochondrial function, which may cause the decrease in ROS, and maintained mitochondrial homeostasis under oxidative stress.

LRPPRC tightly regulates the promotion of cell cycle phases probably by regulating mitochondrial function and mitophagy in physiological and pathological processes. Previous studies have shown that in cancer progression, LRPPRC interacts with Beclin 1 and Bcl-2 and forms a ternary complex to maintain Bcl-2 stability [27,28], resulting in the maintenance of mitochondrial homeostasis and mitochondrial function. By considering that the relative high level of LRPPRC, we employed siRNA targeting to LRPPRC mRNA to modify LRPPRC protein. As expected, knockdown of LRPPRC causes a decrease in Bcl-2, followed by Beclin 1 release to form complexes with PI3KCIII to activate basal levels of autophagy [29]. In our study, we found that under hypoxic conditions, propofol treatment transcriptionally upregulated LRPPRC. Induced HIF-1α transcriptional activity or propofol treatment under normoxic conditions failed to affect LRPPRC, indicating that instead of directly regulating LRPPRC, propofol may exert synergistic effects with a downstream target gene of HIF-1α, which was not detected in this study.

LRPPRC was reported to physically bind to SLIRP and thus was stabilized. Loss of SLIRP impacts mitochondrial gene expression with a slight decrease in mitochondrial DNA mass [23] due to the decrease in LRPPRC resulting in a decrease in mitochondrial membrane potential and mitochondrial dysfunction. By performing immunoprecipitation, we observed that three of these proteins, LRPPRC, SLIRP and Bcl-2, were all observed in both LRPPRC and SLIRP immunoprecipitated products. This indicated that LRPPRC, SLIRP and Bcl-2 may form allotrimer, and potentially regulate stability mutually. These results indicated that LRPPRC, SLIRP and Bcl-2 may form heterotrimers and synergistically promote stabilization and regulate mitochondrial gene expression and mitochondrial function.

In summary, propofol promotes mitochondrial function, maintains mitochondrial homeostasis, inhibits ROS accumulation, and exerts a protective effect against hypoxia-induced oxidative stress, potentially by upregulating LRPPRC. These results provide new insight into the regulatory effects of propofol in mitochondria under oxidative stress.

## Supporting information

**S1 File.**
(ZIP)

## Acknowledgments

The authors apologize to those whose work has not been cited due to space limitations.

## Author Contributions

**Conceptualization:** Qianlu Zhang.

**Data curation:** Qianlu Zhang, Shiwei Cai.

**Funding acquisition:** Guojun Zhao.

**Investigation:** Qianlu Zhang, Liping Guo.

**Methodology:** Qianlu Zhang, Shiwei Cai, Guojun Zhao.

**Resources:** Shiwei Cai.

**Supervision:** Guojun Zhao.

**Writing – original draft:** Liping Guo, Guojun Zhao.

**Writing – review & editing:** Liping Guo, Guojun Zhao.

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
