## [Decision Letter · Decision Letter 0]

4 Jun 2020

PONE-D-20-10919

Mitochondrial-associated protein LRPPRC is critical in propofol-induced protective effect against oxidative stress in cardiac cells

PLOS ONE

Dear Dr. Zhao,

Thank you for submitting your manuscript to PLOS ONE. After careful consideration, we feel that it has merit but does not fully meet PLOS ONE’s publication criteria as it currently stands. Therefore, we invite you to submit a revised version of the manuscript that addresses the points raised during the review process.

Your manuscript was reviewed by two experts and both of them provided major comments. Please address all comments as appropriate.

We look forward to receiving your revised manuscript.

Kind regards,

Partha Mukhopadhyay, Ph.D.

Academic Editor

PLOS ONE

Journal Requirements:

'This work was supported in part by research grants from the National Natural Science Foundation

of China [81601103].'

'The funders had no role in study design, data collection and analysis, decision to

publish, or preparation of the manuscript.'

Additional Editor Comments (if provided):

Reviewers' comments:

Reviewer's Responses to Questions

**Comments to the Author**

1. Is the manuscript technically sound, and do the data support the conclusions?

Reviewer #1: Partly

Reviewer #2: Partly

2. Has the statistical analysis been performed appropriately and rigorously? 

Reviewer #1: Yes

Reviewer #2: Yes

3. Have the authors made all data underlying the findings in their manuscript fully available?

Reviewer #1: Yes

Reviewer #2: Yes

4. Is the manuscript presented in an intelligible fashion and written in standard English?

Reviewer #1: Yes

Reviewer #2: Yes

5. Review Comments to the Author

Reviewer #1: In the current study “Mitochondrial-associated protein LRPPRC is critical in propofol-induced protective effect against oxidative stress in cardiac cells” the authors discussed the involvement of LRPPRC in propoofl-induced cardiac cell protection in hypoxia. Here are some major and minor concerns:

Major:

1. In Figure 1A, why would 48h normoxia group drop in cell viability comparable to the hypoxia group? Does it mean hypoxia is not the reason of cell death?

2. In Figure 1C, pro-caspase-9 and cleaved caspase-9 image is not consistent with statistics.

3. In Figure 2A, without propofol, 24h hypoxia reduces LRPPRC protein expression compared with normoxia group. However, this trend does not show in Figure 2D and many other subsequent figures. The authors need to explain the discrepancy.

4. In Figure 6A, why would hypoxia mock treatment be so different from hypoxia siNC treatment? The increase of LC3-II in hypoxia condition compared to normoxia in Figure 5E is also absent in Figure 6A.

5. The authors found a direct binding between LRPPRC and Bcl-2, and the two protein expression is the same trend. However, it is not sufficient enough to claim that LRPPRC stabilizes Bcl-2. At least the authors need to detect the mRNA Bcl-2, and use translational inhibitors as well as proteasome inhibitors combined with the knockdown of LRPPRC to make a conclusion.

6. Figure 8C right, why would LRPPRC and SLIRP blots not detect the overexpression of these genes?

7. In Figure 8, results claim LRPPRC stabilizes SLIRP, instead of SLIRP stabilizes LRPPRC. This is the opposite of regulation mechanism to the published articles. The authors need to point it out and give a thorough discussion.

8. It is best the authors detect the endogenous SLIRP and LRPPRC binding, because the overexpression system sometimes introduce artificial results due to the expression abundance and overexpression techniques.

Minor:

9. There is no evidence to “Bcl-2 binds and stabilizes the SLIRP protein and thus enhances its function” in the discussion. The authors should be careful in conclusions.

10. Legend of Figure 4 is missing.

11. Instead of LRPPRC protects the mitochondria from autophagy, it is more likely that with LRPPRC, mitochondria are not damaged so that they do not have to be eliminated by autophagy (membrane potential and function is rescued). The authors need to revise the manuscript.

12. It is best the authors put the same genes into groups and distinguish with experimental conditions in the group graphs.

13. The authors never discussed oxidative stress by analyzing ROS or the ROS eliminating mechanisms. The model used is hypoxia and there may be a lot of mechanisms in cytotoxicity. Plus, oxidative stress is best manifested in the reperfusion phase which the authors did not study. So it is best the authors correct the title and related areas from oxidative stress by focusing on hypoxia.

Reviewer #2: The manuscript entitled “Mitochondrial-associated protein LRPPRC is critical in propofol-induced protective effect against oxidative stress in cardiac cells” suggested the protective effects of propofol on hypoxia induced cell injury on H9C2 cells and rat cardiac myocytes via LRPPRC-SLIRP-Bcl2 regulation. In detail, propofol treatment inhibited hypoxia induced apoptosis and autophagy in H9C2 cells. Authors further demonstrated that LRPPRC, SLIRP and Bcl-2 potentially form heterotrimers using immunoprecipitation. Based on accumulated results, the authors clearly described the mechanism of propofol on regulation of mitochondrial homeostasis under hypoxia conditions. However, there are several concerns must be addressed.

1) To support authors’ idea that anti-oxidative stress effects of propofol, oxidative stress markers such as 4HNE or MDA should be checked.

2) In Fig1A, cell viability of normoxia condition at 48h is significantly decreased similar to that of hypoxia condition. Please check the data and if that is correct please discuss it whether that result is related to unchanged expression of LRPPRC with hypoxia induction compared to normoxia group (Fig 2 B, C).

3) For the western blot analysis, please include quantification results. For Fig 1C and 2A, what do the apoptotic death rate graphs mean?

4) For Fig 2A & B, please include normoxia 48h group for control.

5) For Fig 3A, please include HIF-1a western results. The asterisk (*) on results of DFO and CoCl2 treatment groups seems be missing, considering the mention on p16.

6) For Fig 4C, please describe the reason of unchanged HIF-1a protein expression with echinomycin treatment. Please consider for checking HIF-1a expression from nuclear fraction.

7) For Fig 6, data from mock group is confusing. It should be similar to that of siNC.

8) For Fig 6, it should be checked whether maintaining mitochondrial homeostasis effects of propofol against hypoxic conditions was blocked with siLRPPRC transfection.

9) In general, there are some errors/mistakes in the figures. Please re-check them carefully and thoroughly. Some identified was shown below.

- For AnnexinV/PI assay results, indications seem to be incorrect.

Fig 1B & 3C: horizontal: PI/ vertical: Annexin V-> horizontal: Annenxin V/ vertical: PI

Fig4D: missing-> horizontal :PI , vertical : PI

- Fig 4C, please check the indication for propofol treatment +/-.

10) Materials and methods part contains less information.

- Please describe how hypoxia condition was induced on cell cultures.

- Please include methods for siLRPPRC transfection.

- Please add methods for caspase3/9 activity measurements and immunoprecipitation.

6. PLOS authors have the option to publish the peer review history of their article (what does this mean?). If published, this will include your full peer review and any attached files.

Reviewer #1: No

Reviewer #2: No

---

## [Author Response · Author response to Decision Letter 0]

28 Jun 2020

Reviewer #1: In the current study “Mitochondrial-associated protein LRPPRC is critical in propofol-induced protective effect against oxidative stress in cardiac cells” the authors discussed the involvement of LRPPRC in propoofl-induced cardiac cell protection in hypoxia. Here are some major and minor concerns:

Major:

1. In Figure 1A, why would 48h normoxia group drop in cell viability comparable to the hypoxia group? Does it mean hypoxia is not the reason of cell death?

Answer： Dear editor, thanks for the notification. In this section, Normoxia condition with supplemented CoCl2, which is a hypoxia mimic, was considered as a control of hypoxia. By comparing to normoixa+CoCl2 group, the decrease in cell viability by hypoxia exposure was considered to be induced by upregulation of HIF-1a. The figure has been modified.

2. In Figure 1C, pro-caspase-9 and cleaved caspase-9 image is not consistent with statistics.

Answer： Sorry for the mistake. In figure 1C, we incorrectly present the statistical data of image.

3. In Figure 2A, without propofol, 24h hypoxia reduces LRPPRC protein expression compared with normoxia group. However, this trend does not show in Figure 2D and many other subsequent figures. The authors need to explain the discrepancy.

Answer： Thanks for the precious suggestion. According to our results, in figure 2A and C, 24-hour exposure to hypoxia decreased LRPPRC. In figure 2D, 48-hour exposure to hypoxia decreased LRPPRC, but not at 24-hour time point. To figure out the discrepancy, we detected LRPPRC protein level at different time points, including 12, 16, 20, 24, 28, 32, 36, 40, 44, 48h in normoxia and hypoxia group, repeatedly. As shown in our supplemented results (Figure 1), LRPPRC was obviously decreased at 28, 32, 36, 40, and 48 h. However, at 24-hour time point, LRPPRC was not obviously decreased. This instability may explain at 24-h time point after hypoxia exposure, the observation of decrease in LRPPRC was not consistent. That’s why we mainly focused on the 48-hour time point.

Figure 1. After 12, 16, 20, 24, 28, 32, 36, 40, 44 and 48-hour hypoxia exposure,

4. In Figure 6A, why would hypoxia mock treatment be so different from hypoxia siNC treatment? The increase of LC3-II in hypoxia condition compared to normoxia in Figure 5E is also absent in Figure 6A.

Answer：Dear editor, we found that transfection procedure obviously upregulated LRPPRC. This may due to a technical character. If you think it is unacceptable, we’d like to change another technic to introduce siRNA. 

5. The authors found a direct binding between LRPPRC and Bcl-2, and the two protein expression is the same trend. However, it is not sufficient enough to claim that LRPPRC stabilizes Bcl-2. At least the authors need to detect the mRNA Bcl-2, and use translational inhibitors as well as proteasome inhibitors combined with the knockdown of LRPPRC to make a conclusion.

Answer： The stabilizing activity of binding of LRRPRC to Bcl-2 was reported previously (1). We would like to claim that upregulated LRPPRC may function via stabilizes Bcl-2.

Zou J, Yue F, Jiang X, Li W, Yi J, et al. (2013) Mitochondrion-associated protein LRPPRC suppresses the initiation of basal levels of autophagy via enhancing Bcl-2 stability. Biochem J 454: 447-457.

6. Figure 8C right, why would LRPPRC and SLIRP blots not detect the overexpression of these genes?

Answer：Dear editor, in mammal cells, the overexpression of introduced coding sequence is usually not so obvious because of relative low introducing efficacy. The flag- or HA-tagged protein was usually obvious in IP product because of immune enrichment. 

7. In Figure 8, results claim LRPPRC stabilizes SLIRP, instead of SLIRP stabilizes LRPPRC. This is the opposite of regulation mechanism to the published articles. The authors need to point it out and give a thorough discussion.

Answer： Thanks for the suggestion, we also agree with that it is not sufficient to clain LRPPRC stabilizes SLIRP. According to our results in figure 8, we presented the evidence showing that LRPPRC binds to SLIRP only. So we modified the manuscript.

8. It is best the authors detect the endogenous SLIRP and LRPPRC binding, because the overexpression system sometimes introduce artificial results due to the expression abundance and overexpression techniques.

Answer：We previously detected endogenous LRPPRC and SLIRP. However, we failed to intensively detect the binding between endogenous LRPPRC and SLIRP. The potential reasons are listed as followed: (1) antibody target to endogenous LRPPRC and SLIRP are not suitable for immunoprecipitation analysis because of low efficacy. (2) The low endogenous protein level of LRPPRC or SLIRP. 

After several failure, we have to decide to introduce exogenous LRPPRC and SLIRP with Flag or HA tag. 

Minor:

9. There is no evidence to “Bcl-2 binds and stabilizes the SLIRP protein and thus enhances its function” in the discussion. The authors should be careful in conclusions.

Answer： Thanks for the suggestion, it has been modified in 4 th paragraph of discussion section.

10. Legend of Figure 4 is missing.

Answer： Sorry for the mistake!

11. Instead of LRPPRC protects the mitochondria from autophagy, it is more likely that with LRPPRC, mitochondria are not damaged so that they do not have to be eliminated by autophagy (membrane potential and function is rescued). The authors need to revise the manuscript.

Answer： Thanks for the suggestion. We agree with this precious suggestion.

12. It is best the authors put the same genes into groups and distinguish with experimental conditions in the group graphs.

Answer： Do you mean the figure 2A, protein levels?

13. The authors never discussed oxidative stress by analyzing ROS or the ROS eliminating mechanisms. The model used is hypoxia and there may be a lot of mechanisms in cytotoxicity. Plus, oxidative stress is best manifested in the reperfusion phase which the authors did not study. So it is best the authors correct the title and related areas from oxidative stress by focusing on hypoxia.

Answer： Thanks for the suggestion. We modified the title and manuscript accordingly.

Reviewer #2: The manuscript entitled “Mitochondrial-associated protein LRPPRC is critical in propofol-induced protective effect against oxidative stress in cardiac cells” suggested the protective effects of propofol on hypoxia induced cell injury on H9C2 cells and rat cardiac myocytes via LRPPRC-SLIRP-Bcl2 regulation. In detail, propofol treatment inhibited hypoxia induced apoptosis and autophagy in H9C2 cells. Authors further demonstrated that LRPPRC, SLIRP and Bcl-2 potentially form heterotrimers using immunoprecipitation. Based on accumulated results, the authors clearly described the mechanism of propofol on regulation of mitochondrial homeostasis under hypoxia conditions. However, there are several concerns must be addressed.

1) To support authors’ idea that anti-oxidative stress effects of propofol, oxidative stress markers such as 4HNE or MDA should be checked.

Answer： Thanks for the precious suggestion. Instead of 4HNE or MDA, we measured the ROS accumulation, which is also considered as a major cause of oxidative stress. Moreover, we analyzed cell viability and apoptosis as a presentation of oxidative stress-related cell injury. 4HNE and MDA were often employed in patient serum who is suffering oxidative stress-related disease. 

2) In Fig1A, cell viability of normoxia condition at 48h is significantly decreased similar to that of hypoxia condition. Please check the data and if that is correct please discuss it whether that result is related to unchanged expression of LRPPRC with hypoxia induction compared to normoxia group (Fig 2 B, C).

Answer： Dear editor, thanks for the notification. In this section, Normoxia condition with supplemented CoCl2, which is a hypoxia mimic, was considered as a control of hypoxia. By comparing to normoixa+CoCl2 group, the decerase in cell viability by hypoxia exposure was considered to be induced by upregulation of HIF-1a. The figure has been modified.

3) For the western blot analysis, please include quantification results. For Fig 1C and 2A, what do the apoptotic death rate graphs mean?

Answer： Sorry for the incorrect labeling. They have been modified.

4) For Fig 2A & B, please include normoxia 48h group for control.

Answer： The control group was originally included.

5) For Fig 3A, please include HIF-1a western results. The asterisk (*) on results of DFO and CoCl2 treatment groups seems be missing, considering the mention on p16.

Answer： Thanks for the remaindering. The missing part has been added.

6) For Fig 4C, please describe the reason of unchanged HIF-1a protein expression with echinomycin treatment. Please consider for checking HIF-1a expression from nuclear fraction.

Answer： Echinomycin was considered as a inhibitor of HIF-1a’s transcriptional activity. It was proved that echinomycin do not affect HIF-1a protein level.

7) For Fig 6, data from mock group is confusing. It should be similar to that of siNC.

Dear editor, we found that transfection procedure obviously upregulated LRPPRC. This may due to a technical character. If you think it is unacceptable, we’d like to change another technic to introduce siRNA.

8) For Fig 6, it should be checked whether maintaining mitochondrial homeostasis effects of propofol against hypoxic conditions was blocked with siLRPPRC transfection.

Answer： Introduction of siNC obviously upregulated LRPRC, which is unexpected. Knockdown of LRPPRC by siLRPPRC introduction obviously decreased mitochondrial mass.

9) In general, there are some errors/mistakes in the figures. Please re-check them carefully and thoroughly. Some identified was shown below.

- For AnnexinV/PI assay results, indications seem to be incorrect.

Fig 1B & 3C: horizontal: PI/ vertical: Annexin V-> horizontal: Annenxin V/ vertical: PI

Fig4D: missing-> horizontal :PI , vertical : PI

- Fig 4C, please check the indication for propofol treatment +/-.

Answer： Thanks for these suggestions. All figures have been re-checked.

10) Materials and methods part contains less information.

- Please describe how hypoxia condition was induced on cell cultures.

- Please include methods for siLRPPRC transfection.

- Please add methods for caspase3/9 activity measurements and immunoprecipitation.

Answer： Thanks for these suggestions. It has been re-checked.

---

## [Decision Letter · Decision Letter 1]

28 Jul 2020

PONE-D-20-10919R1

Propofol induces Mitochondrial-associated protein LRPPRC and protectives mitochondria against oxidative stress in cardiac cells

PLOS ONE

Dear Dr. Zhao,

Thank you for submitting your manuscript to PLOS ONE. After careful consideration, we feel that it has merit but does not fully meet PLOS ONE’s publication criteria as it currently stands. Therefore, we invite you to submit a revised version of the manuscript that addresses the points raised during the review process.

Your manuscript was reviewed by same reviewers and one reviewer raised some suggestions to improve the quality of the manuscript.

We look forward to receiving your revised manuscript.

Kind regards,

Partha Mukhopadhyay, Ph.D.

Academic Editor

PLOS ONE

Reviewers' comments:

Reviewer's Responses to Questions

**Comments to the Author**

1. If the authors have adequately addressed your comments raised in a previous round of review and you feel that this manuscript is now acceptable for publication, you may indicate that here to bypass the “Comments to the Author” section, enter your conflict of interest statement in the “Confidential to Editor” section, and submit your "Accept" recommendation.

Reviewer #1: (No Response)

Reviewer #2: All comments have been addressed

2. Is the manuscript technically sound, and do the data support the conclusions?

Reviewer #1: Yes

Reviewer #2: (No Response)

3. Has the statistical analysis been performed appropriately and rigorously? 

Reviewer #1: Yes

Reviewer #2: (No Response)

4. Have the authors made all data underlying the findings in their manuscript fully available?

Reviewer #1: Yes

Reviewer #2: (No Response)

5. Is the manuscript presented in an intelligible fashion and written in standard English?

Reviewer #1: Yes

Reviewer #2: (No Response)

6. Review Comments to the Author

Reviewer #1: In the present version of “Propofol induces Mitochondrial-associated protein LRPPRC and protectives mitochondria against oxidative stress in cardiac cells”, the authors made a lot of explanation, correction and improvement. Here are some more minor concerns to be addressed:

1. If Figure 1A is the only experiment to supplement CoCl2 to mimic hypoxia but called control or normoxia, it is best not to show the CoCl2 treatment results. Instead, a non-treated group should be shown to make the whole study consistent. CoCl2 does not provide any more information, but confusion to the study.

2. If LRPPRC expression is modified by siRNA transfection, the authors should mention it in the result or discussion parts to reduce confusion.

3. Although the authors changed the title, oxidative stress is still in it. If the authors insist on claiming oxidative stress, ROS detection should be added and the reperfusion phase should be studied. The source of ROS and ROS elimination machinery should also be analyzed. The easier way is to correct oxidative into hypoxia, which is what the authors studied in the present research project. Also, is it "protectives" or "protects"?

4. For Figure 1C, Figure 2A, the authors should put genes in groups and assign different colors to different experimental condition, the same way of Figure 6D.

5. The authors need to double check the Annexin V/PI staining original figures. It seems in Figure 1 and 3, all the axes for Annexin V is labeled PI and vice versa.

Reviewer #2: (No Response)

7. PLOS authors have the option to publish the peer review history of their article (what does this mean?). If published, this will include your full peer review and any attached files.

Reviewer #1: No

Reviewer #2: No

---

## [Author Response · Author response to Decision Letter 1]

31 Jul 2020

Reviewer #1: In the present version of “Propofol induces Mitochondrial-associated protein LRPPRC and protectives mitochondria against oxidative stress in cardiac cells”, the authors made a lot of explanation, correction and improvement. Here are some more minor concerns to be addressed:

1. If Figure 1A is the only experiment to supplement CoCl2 to mimic hypoxia but called control or normoxia, it is best not to show the CoCl2 treatment results. Instead, a non-treated group should be shown to make the whole study consistent. CoCl2 does not provide any more information, but confusion to the study.

Answer: Thanks for the suggestion. Originally, we already performed normoxia group without cocl2. We’d like to replace normoxia+cocl2 group with normoxia group. 

2. If LRPPRC expression is modified by siRNA transfection, the authors should mention it in the result or discussion parts to reduce confusion.

Answer: It has been mentioned in discussion parts.

3. Although the authors changed the title, oxidative stress is still in it. If the authors insist on claiming oxidative stress, ROS detection should be added and the reperfusion phase should be studied. The source of ROS and ROS elimination machinery should also be analyzed. The easier way is to correct oxidative into hypoxia, which is what the authors studied in the present research project. Also, is it "protectives" or "protects"?

Answer: Thanks for the suggestion. We further modified the title.

4. For Figure 1C, Figure 2A, the authors should put genes in groups and assign different colors to different experimental condition, the same way of Figure 6D.

Answer: Thanks for the suggestion. We further modified the figures.

5. The authors need to double check the Annexin V/PI staining original figures. It seems in Figure 1 and 3, all the axes for Annexin V is labeled PI and vice versa.

Answer: Thanks for the suggestion. We further modified the figures.

---

## [Decision Letter · Decision Letter 2]

26 Aug 2020

Propofol induces Mitochondrial-associated protein LRPPRC and protects mitochondria against hypoxia in cardiac cells

PONE-D-20-10919R2

Dear Dr. Zhao,

We’re pleased to inform you that your manuscript has been judged scientifically suitable for publication and will be formally accepted for publication once it meets all outstanding technical requirements.

Kind regards,

Partha Mukhopadhyay, Ph.D.

Section Editor

PLOS ONE

Additional Editor Comments (optional):

Reviewers' comments:

Reviewer's Responses to Questions

**Comments to the Author**

1. If the authors have adequately addressed your comments raised in a previous round of review and you feel that this manuscript is now acceptable for publication, you may indicate that here to bypass the “Comments to the Author” section, enter your conflict of interest statement in the “Confidential to Editor” section, and submit your "Accept" recommendation.

Reviewer #1: All comments have been addressed

2. Is the manuscript technically sound, and do the data support the conclusions?

Reviewer #1: (No Response)

3. Has the statistical analysis been performed appropriately and rigorously? 

Reviewer #1: (No Response)

4. Have the authors made all data underlying the findings in their manuscript fully available?

Reviewer #1: (No Response)

5. Is the manuscript presented in an intelligible fashion and written in standard English?

Reviewer #1: (No Response)

6. Review Comments to the Author

Reviewer #1: (No Response)

7. PLOS authors have the option to publish the peer review history of their article (what does this mean?). If published, this will include your full peer review and any attached files.

Reviewer #1: No

---

## [Editor Report · Acceptance letter]

28 Aug 2020

PONE-D-20-10919R2 

Propofol induces Mitochondrial-associated protein LRPPRC and protects mitochondria against hypoxia in cardiac cells 

Dear Dr. Zhao:

I'm pleased to inform you that your manuscript has been deemed suitable for publication in PLOS ONE. Congratulations! Your manuscript is now with our production department. 

Kind regards, 

on behalf of

Dr. Partha Mukhopadhyay 

Section Editor

PLOS ONE